# The Influence of Child-Related Factors on Caregiver Perceptions of Their Child’s Sustained Participation in a Community Football Program: A Study of Children with and without Neurodevelopmental Disorders

**DOI:** 10.3390/ijerph18020831

**Published:** 2021-01-19

**Authors:** Carmel Sivaratnam, Bethany Devenish, Tayla Chellew, Nicole Papadopoulos, Jane McGillivray, Nicole Rinehart

**Affiliations:** 1Deakin Child Study Centre, School of Psychology, Faculty of Health, Deakin University, Geelong 3217, Australia; bethany.devenish@deakin.edu.au (B.D.); tayla.chellew@deakin.edu.au (T.C.); nicole.papadopoulos@deakin.edu.au (N.P.); jane.mcgillivray@deakin.edu.au (J.M.); nicole.rinehart@deakin.edu.au (N.R.); 2School of Psychology, Faculty of Health, Deakin University, Geelong 3217, Australia

**Keywords:** participation, involvement, organised physical activity, neurodevelopmental disorders, autism spectrum disorder

## Abstract

This study evaluated the influence of activity preference and involvement on season completion in a community-based football program for children with and without neurodevelopmental disorders. Caregivers (*n* = 1428) of 1529 children aged 4 to 17 (*M* = 7.27, *SD* = 1.85), with (*n* = 175) and without (*n* = 1354) neurodevelopmental disorders who were currently participating or had previously participated in a group-based NAB AFL Auskick football program completed an online survey. The survey collected information on their child’s completion of any attempted seasons of the football program, level of involvement during the sessions and preference for football over other sports and activities. Eighty percent of children with a neurodevelopmental diagnosis had completed all seasons of Auskick, compared with 93% of children without a neurodevelopmental diagnosis. Results indicated that children with neurodevelopmental disorders (*n* = 135) were 3.71 times less likely to complete a football season than their typically developing peers (*n* = 903). Higher levels of involvement during football sessions and greater preference for football were linked to a higher football season completion rate, irrespective of neurodevelopmental disability diagnosis. This study highlights the influence of child-related factors, in particular, preference and involvement, on children’s sustained participation in community football programs, regardless of neurodevelopmental disability status.

## 1. Introduction

Neurodevelopmental disorders (NDDs), including autism spectrum disorder (ASD), attention-deficit/hyperactivity disorder (ADHD), intellectual disability (ID), cerebral palsy (CP) and Down syndrome, are lifelong conditions which vary significantly in nature and severity. They are characterised by cognitive, behavioural, communicative and mobility deficits, which can result in impaired functioning across multiple contexts [1]. Participation in organised physical activity (OPA) programs—that is, formal structured physical activities that are supervised by a coach/adult [2,3] have physical, social and psychological benefits for children with NDDs [2,4,5,6], suggesting that participation in OPA is instrumental in promoting positive health and well-being. Nevertheless, a growing body of literature indicates that children with NDDs participate less frequently in OPA than their typically developing (TD) peers [7,8,9], and are at greater risk for poorer physical and mental health outcomes [10,11,12]. Given OPA participation rates of children with NDDs tend to decline with age [8,13,14], gaining a greater understanding of participation in OPA may be an important step in optimising mental and physical health outcomes for children with NDDs.

Research exploring participation in OPA has found a range of child (e.g., symptom severity), familial (e.g., parent support) and systemic (e.g., program and policy-related) barriers and facilitators for children with NDDs [8,10,15,16,17,18]. Much of this research, however, measures participation as the number of sports a child has attended [10,16] or their frequency of attendance [10,16,18]. This approach to measuring participation assumes that all children wish to participate in OPA, and as argued by Kiuppis [19], this is not always the case, whether they have a disability or not. Instead, it may be important to distinguish between the children and their families who are wanting to participate in OPA and those who are not; prioritising the former when evaluating barriers and facilitators to participation in OPA. Measuring participation as the completion of a sporting season in which a child was enrolled may be one way to address this gap in the existing literature, as it may be more likely to capture children who wanted to take part in OPA and succeeded, or were unable to overcome barriers to participation across the season. For example, qualitative research conducted by Tsai et al. [17] looked at whether children successfully integrated into sports programs, or whether families gave up and withdrew their child.

One framework that may be helpful for understanding OPA participation in children with NDDs in the context of season completion is Imms et al.’s (2017) Family of Participation-Related Constructs (fPRC) framework [20]. This framework delineates the concepts of attendance and involvement as two different constructs coming under the umbrella of participation. Imms et al. define attendance as the amount of activities or the frequency of taking part in activities, and involvement as the “in the moment” experience of participation, including engagement, motivation, persistence, affect (feelings while participating) and social connections [21]. Imms and colleagues further posit that preference, activity competence and sense of self are essential participation-related concepts that influence future participation and are influenced by past and present participation [20,21,22]. Imms et al. [22] further acknowledge the influence of environmental context on participation as described by Maxwell, Alves and Granlund [23], namely availability, accessibility, affordability (financial, time, energy and resources), accommodability and acceptability of activities or services.

When considering participation in OPA in children with NDDs, the influence of involvement has received little attention. Egilson, Jakobsdóttir, Ólafsson and Leósdóttir found that children with ASD were less involved in OPA compared to their TD peers [24]. Additionally, Ryan et al. found that youth with ASD and ID have less positive social experiences in sport compared to youth with only ID [16]. Furthermore, studies have found that children with ASD have significantly less enjoyment for physical activity than their TD peers [25,26], however, these findings are not consistent across the literature, with some research reporting no significant differences in the enjoyment of physical activity when comparing children with ASD [27,28] and children with CP to their TD peers [29]. Research thus far has produced conflicting findings and appears to mainly focus on children with ASD; further research investigating involvement in OPA for children with a diverse range of NDDs is needed. Nevertheless, broader theories such as self-determination theory (SDT) also lend support to the centrality of involvement in participation by highlighting the influence of concepts such as relatedness and autonomy in influencing motivation to participate in exercise [30,31].

The fPRC framework identified child activity preference as another factor influencing participation [20,21,22]. Children with NDDs tend to prefer screen time and other sedentary or individual activities over physical activity [10,15,18]. When investigating children with ASD, Obrusnikova and Cavalier found that 94% of their sample reported that the reason for not participating in physical activity after school was due to engaging in technology-based activities [32]. Similarly, Potvin et al. found that children with high functioning ASD had a lower preference for physical activities compared to their TD peers [28]. When investigating children with Down syndrome, they have been found to be less active than their siblings and prefer indoor activities [33]. Additionally, boys with ADHD have been found to express many negative feelings towards physical activity [34]. While the fPRC suggests that preference may be an important avenue to enhance participation [20,21,22], it seems likely that a lower preference for physical activity may hinder participation in OPA for children with NDDs [35].

The aim of the current study was to investigate the influence of child-related factors, namely activity preference and involvement, on season completion in community-based physical activity in 253 female and 939 male children aged 4 to 17 (*M* = 7.24, *SD* = 1.75) with and without NDDs. This was explored in the context of Australian Rules Football (AFL), one of the most popular sports in Australia [36,37]. NAB AFL Auskick (referred to as Auskick from this point) aims to equip children aged 5 to 12 with motor skills that will support their future participation in AFL and other OPA [38]. Over 200,000 Australian children participate in Auskick, and this number continues to increase each year [39]. Very few studies have explored Auskick participation in children with NDDs. In a preliminary study that qualitatively explored perspectives of children with CP participating in Auskick, child interest in AFL, previous exposure to AFL and the sense of communal identity and belonging were identified by parents and clinicians as important facilitators of participation, whereas barriers identified included game factors, child disability and confidence, and parent apprehension about their child’s abilities [40]. Similarly, parents who participated in interviews as part of an exploratory study of an Auskick program adapted for children with ASD identified that their child’s symptomology, namely social-communication deficits, repetitive interests and sensory difficulties, can act as a barrier to participation [38]. To our knowledge, however, there has been no research to date examining season completion of Auskick for children with NDDs alongside a consideration of factors influencing on season completion. Furthermore, given the large focus in the current literature on the concept of attendance when measuring participation, there is a dearth of knowledge on the influence of activity involvement.

Given that previous research has identified clear links between lower participation in OPA and NDDs [7,8,9], it was expected that season completion would be lower in children with NDDs in comparison to children without NDDs. For the purposes of this study, the term season completion refers to a child’s completion of all attempted seasons of Auskick. This study further aims to evaluate whether failure to complete an Auskick season is related to child involvement in Auskick and child preference for Auskick, and whether this differs between children with NDDs and children without NDDs.

Given that research has found lower levels of involvement in OPA and/or physical activity for children with NDDs [16,24,25,26], it was hypothesised that lower levels of involvement would be found in children with NDDs. It was also predicted that lower levels of involvement would be found in children who had failed to complete a season, regardless of NDD status. Further, given child preferences for sedentary or individual activities has been linked to reduced participation in children with NDDs [10,15,18,28,32], it was hypothesised that a lower preference for Auskick would be found in children with NDDs. Furthermore, it was predicted that lower preference for Auskick would be found in children who had failed to complete a season, regardless of NDD status.

## 2. Materials and Methods

### 2.1. Participants

Participants were 1428 parents or guardians of 1529 children aged 4 to 17 who were currently participating or had previously participated in the Auskick program. To be eligible for enrolment in this study, participants were required to have sufficient English language skills to complete the survey. Participants were recruited through AFL Victoria networks and social media. No participant who consented to participate withdrew from the study.

The characteristics of the participating children and their families are presented in Table 1 and an overview of neurodevelopmental diagnoses is presented in Table 2. Parent respondents were aged 24 to 69 years (*M* = 40.22, *SD* = 6.16). Six percent of families had a combined annual household income of less than $58,188, 36% earned between $58,188 and $105,924, 22% earned more than $105,924, and 35% preferred not to say or did not respond (Note: the Australian median household income for 2017 to 2018 was $88,452). The only significant difference between groups on any demographic variable was that the parent respondents of children with NDDs were significantly more likely to be female than parent respondents of children without NDDs, χ^2^(1, *n* = 681) = 10.82, *p* < 0.001. Parent gender was therefore included in the regression analyses to control for this potentially confounding factor.

### 2.2. Measures

Participants completed (i) a range of demographic questions, in addition to the key measures (ii–v) outlined below. The survey questions were formulated based on the fPRC framework [20,22] and developed by a team of health professionals including clinical psychologists and a physiotherapist who are experts in the field of disability.

(ii) Auskick season completion. Season completion in this study refers to the completion of all attempted seasons of Auskick. Parents were asked to respond yes or no to the question “has your child ever not finished a season of footy?”

(iii) Child involvement in Auskick. Five items using a five-point Likert scale (“does not describe my child” to “describes my child extremely well”) were administered to rate parent’s perceptions of their child’s involvement in Auskick. The five items asked parents whether their child appears motivated to play, persists throughout the activity, feels a social connection, appears to be happy and appears involved in the activity. Responses were averaged, with higher scores indicating increased involvement in Auskick.

(iv) Child preference for Auskick. Four items using a five-point Likert scale (“strongly disagree” to “strongly agree”) were administered to rate parent’s perceptions of their child’s sense of connection to Auskick. The four items asked parents whether Auskick is important to their child, whether Auskick holds special meaning to their child, whether their child prefers Auskick to other organised physical activities, and whether their child prefers Auskick to other activities in general. Responses were averaged, with higher scores indicating a stronger sense of connection.

(v) Neurodevelopmental disability. Six items were administered to measure the presence of neurodevelopmental disability. Parents were asked to indicate, via a list of tick-box options, whether their child had been diagnosed by a health professional with ADHD, autism/ASD/Aspergers syndrome/PDD-NOS, cerebral palsy, dyslexia/dyscalculia, Down syndrome or intellectual disability. Selection of one or more of these options indicated the presence of a neurodevelopmental disability. Parents were also asked to indicate via tick-box options whether their child also had any co-morbid developmental disability diagnoses in addition to the primary diagnosis.

### 2.3. Procedure

Ethics approval was provided by Deakin University Human Research Ethics Committee on 9 August 2016 (DUHREC: 2016-225) and the AFL Research Group. An invitation to participate in the study was emailed to all caregivers on the Auskick database who had provided consent to receiving emails from Auskick. Those who agreed to participate accessed the plain language statement outlining that completion of the survey online indicated they had consented to participate, prior to completing the questionnaire on a survey-based website. Some identifiable data was collected online to enable overall findings to be sent to participants. This identifiable data was stored in password-protected files, and identifying data was removed when the data was collated in a dataset for analysis.

### 2.4. Statistical Analysis

All statistical analyses were conducted using IBM SPSS statistics version 25 (IBM Corp., Armonk, NY, USA). As scales were not included in this study, missing data were deleted list-wise, and variables, where more than 20% of data were missing, were excluded from analyses. Descriptive statistics were conducted to characterise the sample and variables of interest. In order to explore the factors that predict Auskick completion, logistic regression analyses were performed with completion (has not completed a season or has completed all seasons) as the binary dependent variable. Two-way ANOVAs were conducted to explore the effect of the Auskick season completion and neurodevelopmental disability diagnosis on child involvement in and preference for Auskick.

## 3. Results

### 3.1. Participant Characteristics

The parents of 253 female (18%) and 939 male (82%) children aged 4 to 17 (*M* = 7.24, *SD* = 1.75) provided data for Auskick completion, and were included in the analyses. Participants with missing data for Auskick completion were significantly more likely to be from lower income households *t*(1365) = 3.34, *p* < 0.001 and their child was more likely to be female χ^2^(2, *n* = 1724) = 17.24, *p* < 0.001. There were no differences identified for respondent age *t*(1352) = −0.94, *p* = 0.35 or gender χ^2^(2, *n* = 1402) = 2.48, *p* = 0.29.

### 3.2. Auskick Season Completion

Eighty percent of children with NDDs had completed all seasons of Auskick in which they had enrolled, and 93% of children with no NDD diagnosis had completed all enrolled seasons. Auskick season completion was not significantly related to the number of years playing Auskick (*r*(406) = 0.06, *p* = 0.25) or the child’s age (*r*(455) = −0.00, *p* = 0.97), and both these variables had a large amount of missing data. The number of years playing Auskick and child’s age were therefore excluded from the regression analyses. Household income had 15% of responses missing. Little’s Missing Completely at Random test indicated that data were missing at random (χ^2^ = 3.498, df = 2, *p* = 0.17) and so list-wise deletion was used.

The logistic regression analysis showed that a model including parent gender, child gender, household income and child disability status was statistically significant (χ^2^(4) = 28.51, *p <* 0.001) indicating that it differentiated between children who completed all Auskick seasons and children who did not complete a season. The model correctly classified 91.5% of the cases. As shown in Table 3, children with NDDs were 3.71 times more likely to not complete a season of Auskick than children without NDDs (OR = 1.31, 95% CI (2.24, 6.15), *p* < 0.001).

### 3.3. Child Involvement in Auskick

Means and standard deviations for child involvement are presented in Table 4. A two-way ANOVA was conducted to examine whether there were differences in child involvement in Auskick based on Auskick season completion or incompletion and presence or absence of a neurodevelopmental disability diagnosis. There was no statistically significant interaction between Auskick season completion and neurodevelopmental disability diagnosis *F*(1,3) = 1.39, *p* = 0.24. The main effect of Auskick season completion on child involvement in Auskick was significant *F*(1,3) = 5.01, *p* = 0.03. Specifically, those who did complete all seasons of Auskick (*M* = 3.92, *SE* = 0.06) had significantly greater involvement in Auskick than those who did not complete a season of Auskick (*M* = 3.45, *SE* = 0.20). The main effect of neurodevelopmental disability diagnosis was not significant *F*(1,3) = 0.002, *p* = 0.97.

### 3.4. Child Preference for Auskick

Means and standard deviations for child preference for Auskick are presented in Table 5. A two-way ANOVA was conducted to examine whether there were differences in child preference for Auskick based on Auskick season completion or incompletion and presence or absence of a neurodevelopmental disability diagnosis. There was no statistically significant interaction between Auskick season completion and neurodevelopmental disability diagnosis *F*(1,3) = 1.46, *p* = 0.23. The main effect of Auskick completion on child preference for Auskick was significant *F*(1,3) = 17.88, *p* = 0.000. Specifically, those who did complete all seasons of Auskick (*M* = 3.77, *SE* = 0.06) had significantly greater involvement in Auskick than those who did not complete a season of Auskick (*M* = 3.00, *SE* = 0.81). The main effect of neurodevelopmental disability diagnosis was not significant *F*(1,3) = 0.01, *p* = 0.91.

## 4. Discussion

The current study aimed to examine the influence of a diagnosis of NDD, alongside child activity involvement and preference, on season completion in Auskick, a community-based football program for children. The findings showed that children with NDDs were less likely to complete an Auskick season than their peers without NDDs. Higher levels of involvement in Auskick and higher child preference for Auskick were linked to higher rates of Auskick season completion, irrespective of neurodevelopmental disability diagnosis. These results suggest that the lower season completion rates by children with NDDs in comparison to peers are unlikely to be related to a lack of preference for and involvement in Auskick.

In line with previous research findings of lower levels of participation in OPA for children with NDDs [7,8,9], failure to complete an Auskick season was predicted by the presence of a neurodevelopmental disability. The present study extends previous findings by measuring participation as the completion of the attempted Auskick season rather than measuring through frequency of attendance or number of sports engaged in. Measuring failure or success in completing a season may help identify whether there are barriers that prevented children with NDDs and their families from persevering through the season. Simple program adaptations at Auskick can make the program more accessible and make participation more sustainable for children with ASD and their families [38], however, a greater understanding of the specific barriers children with NDDs experience when it comes to successfully completing regular Auskick programs is first needed, prior to proposing any further adaptations.

Children who completed all attempted Auskick seasons had significantly higher levels of involvement in the program than children who failed to complete a season of Auskick, regardless of whether they were diagnosed with NDDs or not. There were no significant differences between children with NDDs and their peers in relation to Auskick involvement. Indeed, of interest is the lack of interaction between NDD and Auskick completion on involvement in Auskick. Given these results, it seems that caregivers of children with NDDs feel their child experiences similar levels of social connection, affect, engagement, motivation and persistence with Auskick as their peers. Therefore, differences in season completion between children with NDDs and their unaffected peers are unlikely to be due to these factors. While results are inconsistent with some initial literature finding significant differences in involvement in OPA between children with and without NDDs [24,25,26], this finding fits with other past studies reporting no significant differences between children with and without NDDs in their enjoyment or intensity (frequency) of participation in physical activity [27,28,29]. In sum, these results provide strong support for an interrelationship between attendance and involvement. They highlight, however, that the ability of children with NDDs to attend or complete a season of Auskick may be significantly disrupted while their involvement is not.

Similarly, children who completed all seasons of Auskick had a greater preference for Auskick compared to children who did not complete a season, regardless of NDD status. There were no differences in preference for Auskick between children with NDDs and their unaffected peers. This conflicts with previous research that found children with NDDs tend to prefer sedentary or individual activities [10,15,18]. One possible explanation for this finding is that all children who were surveyed had been enrolled in Auskick, which may reflect a stronger preference for Auskick to begin with than the broader population of children with NDDs. Further, Australian Rules Football holds a central place in Australian culture and is a topic that is discussed among children during school, even by those who have no interest in playing football [40]. Clinicians and parents of children with CP recognise community culture, identity and sense of belonging as key facilitators to participation in Auskick, as well as the popularity of football within the community and the importance of football in the wider state [40]. Although further investigation is required, it appears that the popularity of AFL across the nation may potentially set Auskick apart from other physical activities for Australian children with NDDs. These results highlight that the ability of children to attend or complete a season of Auskick may be significantly impacted by their preference for Auskick similarly across children with and without NDDs.

The findings from this study are consistent with the fPRC framework proposed by Imms and colleagues [20,22]. Attendance (measured as completion of all attempted Auskick seasons) and involvement were related in both children with and without disabilities, as was attendance with a preference for Auskick. Of note, however, were the findings that children with NDDs were significantly less likely to complete an Auskick season despite experiencing similar levels of involvement and preference for Auskick as their peers, indicating that other factors may be more important for season completion. Imms et al. [20,22] identified activity competence and sense of self as important factors for participation, however, these constructs were not measured in the current study and could possibly account for poorer season completion in children with NDDs. Previous research has identified inadequate program modifications that may act as barriers to the attendance of children with NDDs [16,17], and it seems likely that poor modifications could impact activity competence (as children may be less successful to complete activities without additional supports or changes in an activity), and sense of self (as lowered activity competence will likely impact self-confidence). For example, children with autism tend to have motor skill deficits [41], and children with cerebral palsy experience significant disruptions to their movements [1]. Without program modifications that ensure child success and a sense of competence, children and their families may be more likely to drop out of the season. This premise is supported by research indicating that children with movement deficits experience increasing skill gaps as they grow older thought to be due in part to unmet needs for modified activities and individualised instructions [42].

There does remain, however, a lack of clarity as to why there were no differences in child involvement between children with and without NDDs given we would expect poorer program modification, activity competence and sense of self to impact their engagement, motivation, persistence, affect and social connections. It is possible therefore that the higher rate of season incompletion in children with NDDs may be more closely linked to environmental factors. For example, transportation [32,43], limited financial resources [8,43,44,45] and poor staff–child ratios [18] have been identified as barriers to the participation of children with NDDs in OPAs. Further, the limited availability of accessible and safe facilities and equipment [16,32], poor weather [32] and perceived stigma or negative attitudes from other families can also act as barriers [17,18].

### 4.1. Limitations

There were several limitations to this study. First, the number of children who did not complete a season of Auskick was low, and the overall sample consisted predominantly of children with ASD and/or ADHD, therefore findings should be interpreted with caution, particularly in terms of generalisability to all children with NDDs. Future studies examining the profile of participation and involvement by specific NDD diagnostic group will shed further light on predictors of participation which may be unique to each group. Second, standardised measures were not employed, and the measure of attendance was binomial—simply recording whether they completed all attempted seasons or not. An additional objective measure of season completion (e.g., coach attendance records) in future studies will provide a more robust measurement of attendance. Third, variations across types of sports, specific Auskick programs, geographic locations, specific disorders and age were not accounted for in the current study. Future research could examine whether interactions between involvement and participation in OPA differs at different developmental stages and across different types of sports, and further, whether program characteristics such as resources, geographical accessibility, and coaching approaches and level of training or disorder characteristics such as specific disorders or disorder severity influence child involvement and season completion. Finally, reliance was placed solely on the caregiver report for all variables of interest. It is possible that caregivers of children with NDDs have lowered or different expectations of their child’s involvement in OPA, and that there may indeed be group differences in involvement between children with NDDs and their peers using more objective measures of involvement. Conversely, by relying on the caregiver report, this study is more likely to have measured child involvement relative to other settings, and therefore the level of involvement measured may have factored in a child’s abilities. There is indeed a case for measuring involvement as relative to a child’s abilities to ensure group differences do not simply reflect the presentation of a disability, however, future research would benefit from a more comprehensive approach of measurement—specifically, an inclusion of both objective and well-validated self-report measures that can be completed by children, caregivers and coaches.

### 4.2. Implications

A number of implications can be drawn from the current study. First, the findings that children with NDDs are less likely to complete an Auskick season than their peers, despite similar levels of preference for and involvement in Auskick, highlights the importance of further understanding child and environment-related barriers to season completion. The overall relationship found between Auskick season completion, and child preference for and involvement in Auskick indicates that building child preference and involvement may support Auskick season completion, provided other barriers are also addressed. Simple program adaptations, such as small group activities, increased repetition of new skills, short breaks, and visually-based instructions, can make the program more accessible and more successful for children with ASD and their families [38], however, a greater understanding of the specific barriers children with NDDs experience when it comes to successfully completing Auskick programs is first needed, prior to suggesting any further adaptations. In particular, the potential role of activity competence and sense of self in Auskick participation (both season completion and involvement) should be explored, particularly in relation to program modifications that could address the skill gap between children with and without NDDs, and in identifying environmental barriers such as affordability. Finally, the current research highlights the importance of developing measures of participation that capture both objective and caregiver-reported attendance and involvement in OPA in order to gain a deeper understanding of the factors that influence child participation.

A strength of this study is investigating participation in OPA/Auskick for children across a range of NDDs. When investigating barriers and facilitators to participation, much of the previous literature tends to investigate one type of neurodevelopmental disability at a time [8,10,16,17,18]. By including a range of NDDs, we are able to apply theoretical models (i.e., fPRC) more broadly and apply the findings to community settings, (e.g., coaches and sporting organisations), thereby enhancing the participation of children with all NDDs more efficiently.

## 5. Conclusions

This study provided evidence that children with NDDs are less likely to complete an Auskick season, yet have a similar level of involvement in and preference for Auskick as their peers without NDDs. It is clear that participation, especially in children with disabilities, is a multilevel construct which is not yet fully understood. Future research is needed to identify specific barriers to completion of Auskick for children with NDDs, particularly in relation to activity competence and sense of self. This paper highlights the importance of considering the child-related constructs of both attendance and involvement in OPAs when evaluating participation and associated facilitators and barriers in children with and without NDDs.

## Figures and Tables

**Table 1 ijerph-18-00831-t001:** Participant characteristics.

	*N*	*M (SD)*	Range
Years playing Auskick			
Neurodevelopmental	100	3.27 (1.46)	1–6
Not neurodevelopmental	663	3.37 (1.31)	1–6
Total	763	3.35 (1.33)	1–6
Parent age			
Neurodevelopmental	148	39.93 (6.90)	25–64
Not neurodevelopmental	1108	40.19 (6.09)	24–69
Total	1256	40.16 (6.19)	24–69
Child age			
Neurodevelopmental	165	7.85 (1.83)	5–14
Not neurodevelopmental	1298	7.20 (1.84)	4–17
Total	1463	7.27 (1.85)	4–17
	***N***	**%**	
Female child			
Neurodevelopmental	27	15	
Not neurodevelopmental	242	18	
English primary language			
Neurodevelopmental	150	98	
Not neurodevelopmental	1152	99	
Maternal respondent			
Neurodevelopmental	115	76	
Not neurodevelopmental	780	67	

**Table 2 ijerph-18-00831-t002:** Neurodevelopmental disability characteristics.

Primary Neurodevelopmental Disability Diagnosis	*n* (%)	Comorbidity *n* (%)	Main Comorbidity (%)
ADHD	66 (38)	33 (50)	ASD (45)
Autism	112 (64)	38 (34)	ADHD (27)
Cerebral palsy	7 (4)	2 (28)	ASD (14); ID (14)
Down syndrome	9 (5)	4 (44)	Intellectual disability (44)
Dyslexia/dyscalculia	8 (5)	4 (50)	Both ADHD and ASD (38)
Intellectual disability	28 (16)	20 (71)	ASD (43)

Notes: ASD (Autism Spectrum Disorder); ADHD (Attention-Deficit/Hyperactivity Disorder).

**Table 3 ijerph-18-00831-t003:** Logistic regression analysis of Auskick season completion.

Predictor	B (*SE*)	Wald	Sig	Exp (B)	95% CI
Parent gender is male	−0.47 (0.29)	2.66	0.10	0.62	0.35–1.10
Child gender is female	0.17 (0.32)	0.28	0.60	1.18	0.64–2.19
Income	0.05 (0.08)	0.38	0.54	1.05	0.90–1.21
Child has neurodevelopmental disability	1.31 (0.26)	26.02	<0.001	3.71	2.24–6.15
Constant	−2.84 (0.44)	41.68	<0.001	0.06	

**Table 4 ijerph-18-00831-t004:** Means and standard deviations on child involvement in Auskick.

	*N*	*M (SD)*
Completed all seasons of Auskick		
Neurodevelopmental disability	54	3.79 (0.94)
No neurodevelopmental disability	238	4.05 (0.81)
Did not complete a season of Auskick		
Neurodevelopmental disability	6	3.57 (0.95)
No neurodevelopmental disability	16	3.33 (1.10)

**Table 5 ijerph-18-00831-t005:** Means and standard deviations on child connection to Auskick.

	*N*	*M (SD)*
Completed all seasons of Auskick		
Neurodevelopmental disability	54	3.67 (0.71)
No neurodevelopmental disability	238	3.88 (0.76)
Did not complete a season of Auskick		
Neurodevelopmental disability	6	3.08 (0.74)
No neurodevelopmental disability	16	2.83 (1.08)

## Data Availability

The data presented in this study are available on request from the corresponding author.

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
