# Peer review of "The Influence of Child-Related Factors on Caregiver Perceptions of Their Child’s Sustained Participation in a Community Football Program: A Study of Children with and without Neurodevelopmental Disorders"

_ijerph, 2021, doi:10.3390/ijerph18020831_

Round 1
Reviewer 1 Report
My opinions and assessments regarding the value of the article, below I refer to its subsequent components:
- Title: The title of the article announces that the authors take into account the influence of factors depending on children on their participation in organized sports activities. However, the study only included parents and guardians of children who assessed their children's involvement in participation in the Auskick program. Perhaps it was also necessary to take into account the opinions of older children, e.g. 12-17 years old. In practice, it happens that the opinions of children and parents on the same issue are radically different. If the authors agree with this suggestion, the title of the article should be changed to emphasize that it is about parental views on their children's participation in the Auskick program.
- Abstract: The abstract seems to be precise and concise about the content of the article.
- Introduction: In the introduction section, the authors reviewed the literature on the subject, mainly related to the organized physical activity of children. The introduction does not contain critical remarks in relation to the cited literature, or a new or innovative ideas. Nevertheless, it can be considered adequate to the undertaken research topic.
- Research methodology: The authors have sufficiently described the research context. The selection of research tools also seems to be adequate to the research problem. Some doubts are raised by too wide age range of included children and youth. It can be predicted that a 4-year-old child differs from a 17-year-old adolescent in terms of motivation to play sports or other forms of organized physical activity. It would be advisable to consider this problem depending on the child's belonging to a certain developmental age: preschool period, younger school age, adolescence. Authors should comment on this in the Limitations section. Moreover, the formulation of the main research problem raises doubts, as it seems too simple and obvious (p. 3, lines 115-116: “it was expected that season completion would be lower in children with NDDs in comparison to their unaffected peers”. The authors should agree that this problem wording is not very sophisticated.
- Results: There are no statistical characteristics of Child involvement in Auskick and Child preference for Auskick presented in Results. The presented data suggests that the evaluation of both involvement and preference was based on season completion or incompletion, whereas in Material and Methods it is stated that the assessment of involvement and preference was conducted on the basis on several items asking parents about their children using a five-point Likert scale. What is more, the data suggests “no statistically significant interaction between Auskick season completion and neurodevelopmental disability diagnosis”, which seems contrary to the statement that “children with NDDs were 3.71 times more likely to not complete a season of Auskick than children without NDDs”. Therefore I would suggest to present the results in a more clear way. Interpretation of research results includes several doubts that should be clarified. As the title of the article shows, the main dependent variable is "the participation of children in sports activities". In this context, the participants’ involvement in playing sports is an independent variable that can influence the dependent variable in a specific way. However, on p. 8, lines 225-226, the authors state that "The main effect of Auskick season completion on child involvement in Auskick 226 was significant F (1,3) = 5.01, p = .03". This suggests that the dependent variable affects the independent variable in some way, which is a misconception. The authors are asked to clear it.
- Discussion: A significant part of the discussion relates to similar research by other authors. I would suggest the authors to formulate a few proposals for the practice, the implementation of which would contribute to the improvement of children's participation in organized physical activity.
- Conclusions: In my opinion, the conclusions are based on the obtained results.
- Limitations: In addition to the limitations contained in this section, also others should be added, resulting from the above comments and suggestions.
Overall assessment: The reviewed article refers to an issue that is important from a practical point of view, however, the obtained results do not add much to the literature relating to the organized physical activity of disabled children and youth. The article contains data that in one form or another repeat data that can be found in the literature on the subject. For these reasons, the text is not very innovative. On this basis, I do not recommend the Editor to publish a peer-reviewed manuscript.
Author Response
Dear reviewer,
Thank you for taking the time to review and provide feedback on our manuscript. Your feedback provided the authorship team with an opportunity to make further revisions, which we feel has improved the clarity and depth of the manuscript. Our detailed responses to your comments can be found below.
|
Reviewer comments |
Response |
|
Reviewer 1 |
|
|
The title of the article announces that the authors take into account the influence of factors depending on children on their participation in organized sports activities. However, the study only included parents and guardians of children who assessed their children's involvement in participation in the Auskick program. Perhaps it was also necessary to take into account the opinions of older children, e.g. 12-17 years old. In practice, it happens that the opinions of children and parents on the same issue are radically different. If the authors agree with this suggestion, the title of the article should be changed to emphasize that it is about parental views on their children's participation in the Auskick program. |
Thank you for drawing our attention to this – we have updated the title to:
The influence of child-related factors on caregiver perceptions of their child’s sustained participation in a community football program: A study of children with and without neurodevelopmental disorders |
|
Some doubts are raised by too wide age range of included children and youth. It can be predicted that a 4-year-old child differs from a 17-year-old adolescent in terms of motivation to play sports or other forms of organized physical activity. It would be advisable to consider this problem depending on the child's belonging to a certain developmental age: preschool period, younger school age, adolescence. Authors should comment on this in the Limitations section. |
Thank you - we acknowledge that there is potential for variations across age, and have added this into the limitations as suggested on Lines 369-375: “Third, variations across types of sports, specific Auskick programs, geographic locations, specific disorders and age were not accounted for in the current study. Future research could examine whether interactions between involvement and participation in OPA differs at different developmental stages and across different types of sports, and further, whether program characteristics such as resources, geographical accessibility, and coaching approaches and level of training, or disorder characteristics such as specific disorders or disorder severity influence child involvement and season completion” |
|
Moreover, the formulation of the main research problem raises doubts, as it seems too simple and obvious (p. 3, lines 115-116: “it was expected that season completion would be lower in children with NDDs in comparison to their unaffected peers”. The authors should agree that this problem wording is not very sophisticated. |
Thank you for this point. For further clarity, this study had 3 research questions: 1. Does NDD diagnosis predict season completion 2. Does failure to complete Auskick season by children with NDDs relate to lower involvement 3. Does failure to complete Auskick season by children with NDDs relate to lower preference for the sport. “it was expected that season completion would be lower in children with NDDs in comparison to children without NDDs” |
|
Results: There are no statistical characteristics of Child involvement in Auskick and Child preference for Auskick presented in Results. |
Tables 4 & 5 (pages 11 & 12) presents statistical characteristics of child involvement and preference. |
|
The presented data suggests that the evaluation of both involvement and preference was based on season completion or incompletion, whereas in Material and Methods it is stated that the assessment of involvement and preference was conducted on the basis on several items asking parents about their children using a five-point Likert scale. |
Thank you for this point; to add further clarity to how we have come to this presentation of data, the aim of this paper was to evaluate whether two variables (1. Preference for Auskick; 2. Involvement in Auskick sessions) interacted with a third variable (whether children completed an Auskick season) (for outline of aim see lines 99-101). As such, season completion or incompletion was measured as a binomial outcome (see lines 165-167), while preference and involvement were completed on 5-point Likert scales (see lines 168-173 and lines 174-179). In all result sections these variables are identified as separate variables/constructs, and were reported in line with common reporting expectations for two-way ANOVA analyses. |
|
What is more, the data suggests “no statistically significant interaction between Auskick season completion and neurodevelopmental disability diagnosis”, which seems contrary to the statement that “children with NDDs were 3.71 times more likely to not complete a season of Auskick than children without NDDs”. Therefore I would suggest to present the results in a more clear way. |
Thank you for this comment. To provide further clarification, to align with APA and common reporting expectations for ANOVA analyses (see https://statistics.laerd.com/spss-tutorials/two-way-anova-using-spss-statistics-2.php or associated guide) we reported our data using specific terminology (such as interaction, main effects, and reporting the odds ratio). A statement was made as to the nature of the test being conducted immediately prior to the statement about the interaction (“A two-way ANOVA was conducted to examine whether there were differences in child involvement in Auskick based on Auskick season completion or incompletion and presence or absence of a neurodevelopmental disability diagnosis.” Lines 252-254). The statement about the interaction following this indicates, in line with normal ANOVA reporting, that there were no differences in involvement identified at any level of season completion or neurodevelopmental diagnosis. This does not sit at odds with the statement that children with NDDs are more likely to not complete a season, as that statement has a different focus (i.e. it is not looking at how these interact with involvement). Lines 283-287 and 299-307 provide the interpretation of these results in language that makes the meaning clear to readers who are unfamiliar with the statistical analysis, and we feel that offering interpretation within the results section, or reporting the results in a manner that does not align with normal reporting of ANOVA analyses, would not be considered best practice. |
|
Interpretation of research results includes several doubts that should be clarified. As the title of the article shows, the main dependent variable is “the participation of children in sports activities”. In this context, the participants’ involvement in playing sports is an independent variable that can influence the dependent variable in a specific way. However, on p. 8, lines 225-226, the authors state that “The main effect of Auskick season completion on child involvement in Auskick 226 was significant F (1,3) = 5.01, p = .03”. This suggests that the dependent variable affects the independent variable in some way, which is a misconception. The authors are asked to clear it. |
Thank you for providing this feedback. To provide some further explanation as to why we included this interpretation/wording – an ANOVA identifies whether there are significant differences between the means of different variables. So, in our study, we tested whether scores on involvement differed depending on whether they had previously completed all seasons or had been diagnosed. This did not indicate directionality – instead, it shows that (using the example provided re: the main effect of Auskick) children who had completed all seasons of Auskick were more likely to have been rated as more involved during Auskick sessions, than children who had not completed all seasons. Given the survey was cross-sectional, no intervention/manipulation of variables occurred, and the analysis was testing group differences, it is equally acceptable to state either “children who are rated as more involved in Auskick were more likely to complete all seasons” or “children who completed all seasons were more likely to have higher involvement in Auskick sessions”. |
|
Discussion: A significant part of the discussion relates to similar research by other authors. I would suggest the authors to formulate a few proposals for the practice, the implementation of which would contribute to the improvement of children’s participation in organized physical activity.
|
Thank you for this feedback - we have added specific program adaptations that may improve child participation (lines 393-394): “Simple program adaptations, such as small group activities, increased repetition of new skills, short breaks, and visually-based instructions, can make the program more accessible and more successful for children with ASD and their families [39], however a greater understanding of the specific barriers children with NDDs experience when it comes to successfully completing Auskick programs is first needed, prior to suggesting any further adaptations.” |
Reviewer 2 Report
The article is well written with proper grammar and specific scientific expressions, and in my opinion, it meets the journal's editing requirements.
I propose that in the title the term "community sports" be changed to "Auskick program" since during the work only reference is made to this specialty.
In the abstract, you can enter the sample of parents and caregivers who responded to the questionnaire, not just the children studied. In the results mentioned in the abstract, I would include some quantitative data and, in the conclusion, it would not refer to community sports but specifically to Auskick.
The introduction is clear, it includes the fundamental aspects of the study, but it should include the participants indicated in lines 178-179. The introduction focuses on the fact that insufficient attention has been paid to the influence of the environment in children with NDDs who perform physical exercise or to the inconsistency of other studies.
The aim of the study is clear since few studies have described the characteristics of the environment in children with NDDs participating in Auskick.
Material and Method: Participants.
The statistical analysis is adequate for the objectives set with the observations that the same authors indicate in the limitations section.
This section should include the 1,729 parents and caregivers and participating children. Demographic data should also be included in the Participants section (table 1), since it is not about results, but about a description of the sample. And likewise, Table 2 should be added to participants, not to results.
This is an important sample, but being only Auskick practitioners it can be too specific.
As for the measurements, they can number the surveys carried out (1. Demographic questions, 137 line; 2.- 141 line; 3.- 144; 4.- 150 line; 5.- 156 line).
Regarding the Procedure, in addition to the ethical procedures, they should indicate more specifically the circumstances, chronology or way of supplying the surveys to the participants, the system of confidentiality of the data and the preservation of the privacy of the participants.
This article is very interesting because it addresses a socially very relevant problem such as the integration of children with neurodevelopmental problems in conventional physical activities, in this case Auskick.
The discussion focuses on the most relevant aspects of the results and the conclusions are clear and can be applied in this area of knowledge.
The methodology is adequate for this type of study. However, it has limitations in addition to those indicated by the authors in the document. An important limitation of the study is that it does not contemplate the influence of the Auskick programs in the different environments in which the study was carried out, including the role of the coaches, schedules, materials and different methods that could justify a greater or lesser adherence of children to activity. In future researches, it would be necessary to expand the methodology regarding the collection of information in order to be more specific and to be able to determine more clearly that participation in OPA (not only Auskick) allows optimizing mental development and physical health for children with NDDs.
Another aspect to consider is to extend the sample to other sports and to other geographical areas, as well as to focus on the possibility of addressing a single disorder in order to more clearly establish the improvement in children with NDDs and those who practice a specific sport.
The possibility of asking children directly through questionnaires validated according to their disorder should also be considered in order to establish their predisposition instead of focusing only on the perception of parents and caregivers.
The bibliography is current and relevant and is adequately referenced in the document.
The dissemination of this study and application of its results in their environment can serve to promote the integration of children with neurological development deficiencies in the Auskick among parents and groups.
Author Response
Dear reviewer,
Thank you for taking the time to review and provide feedback on our manuscript. Your feedback provided the authorship team with an opportunity to make further revisions, which we feel has improved the clarity and depth of the manuscript. Our detailed responses to your comments can be found below.
|
Reviewer comments |
Response |
|
|
|
|
I propose that in the title the term "community sports" be changed to "Auskick program" since during the work only reference is made to this specialty. |
Thank you. The term ‘Auskick’ may not be used consistently across states and would not be understood globally, and so to ensure most readers would have an understanding of the content from the title we have updated it to:
The influence of child-related factors on caregiver perceptions of their child’s sustained participation in a community football program: A study of children with and without neurodevelopmental disorders |
|
In the abstract, you can enter the sample of parents and caregivers who responded to the questionnaire, not just the children studied. |
Thank you - we have now corrected this (lines 16-17), and included those for whom there was appropriate data in the results section of the abstract instead (Lines 22-24). |
|
In the results mentioned in the abstract, I would include some quantitative data and, in the conclusion, it would not refer to community sports but specifically to Auskick. |
We have expanded to include statistics for percentage of children with and without NDDs who had completed the season (lines 22-23), and added the odds ratio (line 24). Further, we corrected community sports to Auskick as suggested (line 29). |
|
The introduction is clear, it includes the fundamental aspects of the study, but it should include the participants indicated in lines 178-179. |
We have expanded the aim to include the participants (line 101): The aim of the current study was to investigate the influence of child-related factors, namely activity preference and involvement, on season completion in community-based physical activity in 253 female and 939 male children aged 4-17 (M = 7.24, SD = 1.75) with and without NDDs. |
|
Material and Method: Participants This section should include the 1,729 parents and caregivers and participating children. Demographic data should also be included in the Participants section (table 1), since it is not about results, but about a description of the sample. And likewise, Table 2 should be added to participants, not to results. |
Thank you for this feedback - we have made the following changes:
There was an error in the number of caregivers reported originally. Many caregivers responded to survey items for all of their children, including a small number of siblings who had never participated in Auskick. As siblings who have never participated were not included in the present study as they did not meet criteria, their data was excluded. We have corrected the number of caregivers in the abstract (lines 16-17) and the results (line 136). Further, we updated participant demographics to reflect the overall sample (Tables 1 & 2 and lines 142-151), and moved these tables to the methods as recommended (with the exception of Auskick completion – as this was a variable reported in results, we limited this data to the analysis sample, and reported in text lines 234-235). |
|
As for the measurements, they can number the surveys carried out (1. Demographic questions, 137 line; 2.- 141 line; 3.- 144; 4.- 150 line; 5.- 156 line). |
Thank you - we have numbered all key variables as suggested, using roman numerals for flow (lines 162-181) |
|
Regarding the Procedure, in addition to the ethical procedures, they should indicate more specifically the circumstances, chronology or way of supplying the surveys to the participants, the system of confidentiality of the data and the preservation of the privacy of the participants. |
Thank you for drawing our attention to this - we have added the following details (lines 189-197): - ethics approval date and code - information about invitation to participants - information about identifiable data |
|
An important limitation of the study is that it does not contemplate the influence of the Auskick programs in the different environments in which the study was carried out, including the role of the coaches, schedules, materials and different methods that could justify a greater or lesser adherence of children to activity. In future researches, it would be necessary to expand the methodology regarding the collection of information in order to be more specific and to be able to determine more clearly that participation in OPA (not only Auskick) allows optimizing mental development and physical health for children with NDDs. Another aspect to consider is to extend the sample to other sports and to other geographical areas, as well as to focus on the possibility of addressing a single disorder in order to more clearly establish the improvement in children with NDDs and those who practice a specific sport. |
Thank you for noting this. We have added the following text as a limitation in lines 369-375: Third, variations across types of sports, specific Auskick programs, geographic locations, specific disorders and age were not accounted for in the current study. Future research could examine whether interactions between involvement and participation in OPA differs at different developmental stages and across different types of sports, and further, whether program characteristics such as resources, geographical accessibility, and coaching approaches and level of training, or disorder characteristics such as specific disorders or disorder severity influence child involvement and season completion |
|
|
|
|
The possibility of asking children directly through questionnaires validated according to their disorder should also be considered in order to establish their predisposition instead of focusing only on the perception of parents and caregivers. |
We agree, and have incorporated into the following (lines 384-386): …however future research would benefit from a more comprehensive approach of measurement – specifically, an inclusion of both objective and well-validated self-report measures that can be completed by children, caregivers, and coaches. |
|
|
|